# Long-Term Safety in HBsAg-Negative, HBcAb-Positive Patients with Rheumatic Diseases Receiving Maintained Steroid Therapy after Pulse Therapy

**DOI:** 10.3390/jcm10153296

**Published:** 2021-07-26

**Authors:** Ying-Cheng Lin, Yen-Ju Chen, Shou-Wu Lee, Teng-Yu Lee, Yi-Hsing Chen, Wen-Nan Huang, Sheng-Shun Yang, Yi-Ming Chen

**Affiliations:** 1Division of Gastroenterology and Hepatology, Department of Internal Medicine, Taichung Veterans General Hospital, Taichung 40705, Taiwan; ethankevin516@vghtc.gov.tw (Y.-C.L.); ericest@vghtc.gov.tw (S.-W.L.); tylee@vghtc.gov.tw (T.-Y.L.); 2Institute of Clinical Medicine, National Yang Ming Chiao Tung University, Taipei 11221, Taiwan; aoaichen@vghtc.gov.tw; 3Department of Medical Research, Taichung Veterans General Hospital, Taichung 40705, Taiwan; 4Division of Allergy, Immunology and Rheumatology, Department of Internal Medicine, Taichung Veterans General Hospital, Taichung 40705, Taiwan; dr.yihsing@gmail.com (Y.-H.C.); gtim5555@yahoo.com (W.-N.H.); 5School of Medicine, Chung Shan Medical University, Taichung 40201, Taiwan; 6School of Medicine, College of Medicine, National Yang Ming Chiao Tung University, Taipei 11221, Taiwan; 7Rong Hsing Research Center for Translational Medicine, National Chung Hsing University, Taichung 40227, Taiwan; 8Program in Translational Medicine, National Chung Hsing University, Taichung 40227, Taiwan; 9Institute of Biomedical Sciences, National Chung Hsing University, Taichung 40227, Taiwan

**Keywords:** glucocorticoid pulse therapy, hepatitis B surface antigen, seroreversion, rheumatic diseases

## Abstract

The risk of hepatitis B virus (HBV) reactivation in hepatitis B surface antigen (HBsAg)-negative, antibody to hepatitis B core antigen (anti-HBc)-positive patients after glucocorticoid (GC) pulse therapy remains unclear. Aims: Our study aimed to examine the safety of GC pulse therapy in HBsAg-negative, anti-HBc-positive rheumatic patients. Methods: Medical records of HBsAg-negative, anti-HBc-positive patients receiving GC pulse therapy to treat rheumatic diseases were reviewed. The primary outcome was HBV-associated hepatitis occurring within the first year after GC pulse therapy; the secondary outcome was HBsAg seroreversion occurring during the follow-up period. Results: We identified 5222 HBsAg-negative, anti-HBc-positive patients with rheumatic diseases who had attended Taichung Veterans General Hospital from October 2006 to December 2018. A total of 689 patients had received GC pulse therapy, with 424 patients being analyzed. Hepatitis was noted in 28 patients (6.6%) within the first year after GC pulse therapy, but none had been diagnosed as HBV-associated hepatitis. Three patients (0.7%) later developed HBsAg seroreversion, with a median interval of 97 months from the first episode of GC pulse therapy. These cases concurrently had maintained high dose oral prednisolone (≥20 mg prednisolone daily for over 4 weeks). Conclusions: Amongst the HBsAg-negative, anti-HBc-positive rheumatic patients treated with GC pulse therapy, the risk of HBV-associated hepatitis within the first year was low. HBsAg seroreversion may have developed in the later stage, but only in those patients who had maintained high-dose oral steroid.

## 1. Introduction

Despite the wide availability of both vaccines and highly potent antiviral therapy, hepatitis B virus (HBV) infection remains the leading cause of chronic liver disease and hepatocellular carcinoma worldwide [1]. It is estimated that more than one-third of the world’s population has been infected with HBV [2] and that at least 257 million people are chronically infected (defined as the appearance of hepatitis B surface antigen (HBsAg)-positive for longer than 6 months) [1]. In chronically infected patients, HBsAg loss (from HBsAg-positive to HBsAg-negative) may occur either spontaneously or after antiviral therapy. Loss of HBsAg is now considered as the functional cure for chronic hepatitis B (CHB). However, HBV reactivation (defined as the reappearance of HBsAg from negative to positive, or a rise in serum HBV-DNA compared with baseline levels or detection of HBV-DNA if undetectable at baseline) can occur in HBsAg-negative patients, particularly in those who have received immunosuppressive agents [3,4,5]. Amongst the immunosuppressive agents, glucocorticoid (GC) was one of the earliest and most widely-used medications found to precipitate hepatitis B reactivation [6]. Four to 6 weeks of high-dose oral GC therapy (prednisolone equivalent 20 mg to 60 mg daily) resulted in 43% to 67% of HBV reactivation in HBsAg-positive patients [7,8]. Our previous study revealed a hepatitis B virus reactivation rate of 15.3% (11/72) in HBsAg-positive patients who had been treated with intravenous (IV) GC pulse therapy (625 mg to 15,000 mg prednisolone equivalent for one to three days) [9]. However, there is little information in the available literature concerning the frequency or risk of HBV seroreversion in HBsAg-negative, antibody to hepatitis B core antigen (anti-HBc)-positive patients after GC pulse therapy. Our study aimed to investigate the risk of HBV reactivation following large-dose GC treatment in patients with autoimmune rheumatic diseases.

## 2. Materials and Methods

### 2.1. Study Design and Patient Population

Consecutive patients who tested positive for anti-HBc and negative for HBsAg, and had received GC pulse therapy for rheumatologic indications from October 2006 to December 2018 were retrospectively collected at Taichung Veterans General Hospital, Taiwan. Alanine aminotransferase (ALT) was regularly monitored every 3–4 months after GC pulse therapy and HBsAg was checked annually. The International Classification of Diseases, 9th Revision, Clinical Modification (ICD-9-CM) codes and ICD-10-CM codes were used to identify rheumatic diseases of rheumatoid arthritis (ICD-9-CM 714.0 or 714.30–714.33; ICD-10-CM M05.7-M05.9, M06.0, M06.2-M06.3, M06.8-M06.9, or M08), systemic lupus erythematosus (ICD-9-CM 710.0; ICD-10-CM M32), Sjogren’s syndrome (ICD-9-CM 710.2; ICD-10-CM M35.0), ankylosing spondylitis (ICD-9-CM 720.0; ICD-10-CM M45 or M08), psoriatic arthritis (ICD-9-CM 696.0; ICD-10-CM L40.5), idiopathic inflammatory myopathy (ICD-9-CM 710.3 or 710.4; ICD-10-CM M33.0, M33.1, M33.2, M33.9, or M36.0), adult-onset Still’s disease (ICD-9-CM 714.2; ICD-10-CM M06.1), systemic sclerosis (ICD-9-CM 710.1; ICD-10-CM M34), and vasculitis (ICD-9-CM 446.0, 446.4, 446.5, or 446.7; ICD-10-CM M30.1 or M31.3-7). Patients were excluded if they were younger than 20 years of age, had undergone solid organ/hematologic transplant, had active malignancies, liver cirrhosis, tested positive for hepatitis C virus infection, had expired, were lost during follow-up within 3 months after GC pulse therapy or were without follow-up for HBsAg. Patients with a concurrent use of rituximab were also excluded.

### 2.2. Data Collection

#### 2.2.1. Baseline Characteristics

Baseline characteristics of the enrolled individuals included age, gender, alanine aminotransferase (ALT), total bilirubin, albumin, platelet count, and the diagnosis of rheumatic diseases. The number of pulse therapy episodes and the dosages of maintained oral steroids for each individual were also recorded. Other immunosuppressive agents, including both conventional and biological disease-modifying antirheumatic drugs, concomitantly used within 1 year before or after the use of GC pulse therapy were also documented.

#### 2.2.2. Serological and Virologic Tests for HBV

HBsAg, anti-HBc and antibody to hepatitis B surface antigen (anti-HBs) were detected by electrochemiluminescence immunoassay (Roche Diagnostics, Mannheim, Germany). HBV-DNA viral loads were quantified by Roche Cobas TaqMan HBV Test (Roche Diagnostics, Basel, Switzerland).

#### 2.2.3. Glucocorticoid Therapy and Concurrent Medications

GC pulse therapy was administered due to clinical and/or humoral activity or relapse of the underlying diseases. It was defined as treatment involving more than 250 mg prednisone or equivalent per day, for one or more days [10].

Maintained high dose oral steroid, described as prednisolone-equivalent, is defined as the daily dose of maintained GC treatment for more than 4 weeks following the use of GC pulse therapy.

Conventional disease-modifying anti-rheumatic drugs, including methotrexate, azathioprine, leflunomide, cyclosporine, sulfasalazine, mycophenolate, and cyclophosphamide, were recorded. Biological disease-modifying anti-rheumatic drugs of anti-tumor necrosis factor-α (adalimumab, etanercept, golimumab), abatacept, and rituximab were also included in the analysis.

### 2.3. Outcome Definition

HBV-associated hepatitis was defined as an ALT increase 3 times from baseline and >100 U/L, combined with a detectable level of HBV DNA and HBsAg seroreversion [3,4]. HBsAg seroreversion was defined as the reappearance of HBsAg from negative to positive.

Drug-induced hepatitis was defined on the basis of a temporal relationship and the exclusion of alternative explanations, i.e., acute viral hepatitis other than HBV, alcoholic hepatitis or infectious liver diseases [11].

### 2.4. Statistical Analyses

Categorical variables are presented with numbers (percentages) and were compared by the Chi-Square test. Continuous variables are presented with median (interquartile range) and were compared using the Fisher’s exact test. *p* values < 0.05 were considered statistically significant. Multivariate logistic regression analysis was used to evaluate the odds ratio (OR) and 95% confidence interval (CI) of risk factors for hepatitis. Statistical tests were performed using the Statistical Package for the Social Sciences (IBM SPSS version 22.0; International Business Machines Corp, New York, NY, USA).

## 3. Results

### 3.1. Enrolled Patients

The flowchart of patient recruitment is shown in Figure 1. In total, 5222 HBsAg-negative, anti-HBc-positive rheumatic patients were identified. Amongst them, 689 patients received GC pulse therapy, exclusively methylprednisolone 500 mg for 3 days. After excluding patients younger than 20 years, those with concurrent comorbidities, those who were followed up less than 3 months, those without follow-up HBsAg, and those with concurrent use of rituximab, a remaining 424 patients were analyzed. The demographic characteristics are shown in Table 1.

### 3.2. The Occurrence of Hepatitis within the First Year after GC Pulse Therapy

Hepatitis was documented in 28 patients (6.6%) within the first year after GC pulse therapy. Neither a detectable HBV DNA level nor HBsAg seroreversion occurred at the time of hepatitis. After evaluation, the cause of hepatitis was mostly drug-induced. Amongst these subjects with hepatitis, the median ALT level at peak was 157 IU/mL (range 105–939 IU/mL). The median time from GC pulse therapy to hepatitis was 85 days (range 3–365 days). All cases were managed conservatively and were clinically stable without experiencing progression to liver failure. Further analysis of the individuals with hepatitis after GC pulse therapy and those without is shown in Table 2. The patients with hepatitis were older in age (median age 55.2 vs. 46.4, *p* = 0.047), and had a higher baseline total bilirubin level (0.7 vs. 0.4 mg/dL, *p* = 0.006). A higher proportion of patients with hepatitis concurrently used azathioprine (57.1% vs. 34.3%, *p* = 0.026) and mycophenolate (21.4% vs. 7.0%, *p* = 0.018), while a lower proportion of patients were treated with methotrexate (28.6% vs. 56.6%, *p* = 0.007) and sulfasalazine (14.3% vs. 40.7%, *p* = 0.010). As shown in Table 3, after logistic regression analysis, older age (OR: 1.04, 95% CI: 1.01–1.07, *p* = 0.021) and azathioprine use (OR: 2.50, 95% CI: 1.03–6.06, *p* = 0.043) were independent factors associated with hepatitis.

### 3.3. The Ratio of HBsAg Seroreversion after GC Pulse Therapy

As shown in Figure 1, three patients (0.7%), all females, later experienced HBsAg seroreversion during follow-up, with the detailed information listed in Table 4. The median time from the first episode of GC pulse therapy to HBsAg seroreversion was 97 months (range 66–168 months). All of the seroreverters concurrently had maintained oral glucocorticoid higher than 20 mg daily prednisolone equivalent dose for more than 4 weeks. Two of them had a two-fold increase in ALT level at the time of seroreversion. All patients had an abrupt rise in HBV DNA levels. None of the patients had an elevated total bilirubin level, nor any evidence of liver failure after timely treatment with antiviral drugs, exclusively entecavir.

## 4. Discussion

This study demonstrated that no HBsAg-negative, anti-HBc-positive rheumatic patient developed HBV-associated hepatitis within the first year after GC pulse therapy with three cases (0.7%) experiencing HBsAg seroreversion in the later stage. This study is the first to investigate the prevalence of HBV-associated hepatitis and HBsAg seroreversion in HBsAg-negative, anti-HBc-positive rheumatic patients undergoing ultra-high dose, GC pulse therapy. This result indicates that GC pulse could be administered safely in this group of patients with autoimmune rheumatic diseases. According to previous studies, HBV-associated hepatitis is a well-documented complication in HBsAg-negative, anti-HBc-positive patients receiving steroid-containing cytotoxic chemotherapy in lymphoma patients [12,13,14]. In those studies, most HBV-associated hepatitis occurred within the first year after cytotoxic agents. In our current study, 28 cases (6.6%) experienced a hepatitis flare within the first year after steroid pulse therapy. Rather than being HBV-associated hepatitis, these cases were attributable to drug-induced liver injury on the basis of temporal relationship and the exclusion of alternative explanations, i.e., acute viral hepatitis other than HBV, alcoholic hepatitis or infectious liver diseases [11]. Moreover, we found that the risk for hepatitis was not related to either the episodes of pulse steroid or the dosage of maintained oral steroids. Alternatively, it was related to older age and the concurrent use of azathioprine. These findings are similar to previous reports taken in different clinical settings [11,15,16]. One large prospective study which enrolled 1218 subjects in Italy, found that the higher prevalence of hepatitis seen in HBsAg-negative, anti-HBc-positive patients was associated with older age, but not with concomitant immunosuppressants [15]. Moreover, hepatitis is a common side effect of azathioprine in rheumatic patients, as well as in patients who have undergone renal transplantation [11,17]. The mean annual azathioprine-induced liver injury rate has been estimated to be 1.4% [11]. Therefore, it is recommended that a thorough work-up, including HBV DNA level and HBsAg, be performed, in order to distinguish the causes of hepatitis in HBsAg-negative, anti-HBc-positive patients with concurrent azathioprine therapy. The long-term risk of HBsAg seroreversion, with or without hepatitis, has been previously reported in HBsAg-negative, anti-HBc-positive patients receiving systemic glucocorticoids. However, the dosage and duration of systemic steroids that pose the highest threat remains unclear. One large retrospective study performed in Hong-Kong revealed that the 10-year incidence rate of HBsAg seroreversion was 5.5% in patients with previous hepatitis B virus exposure who had received at least one dose of systemic GC for all indications [18]. In this study, both the dosage and duration of systemic steroid resulted in no significant difference in relation to HBsAg seroreversion. A retrospective study from Taiwan demonstrated that HBsAg seroreversion can occur 120 months after steroid exposure (range 20–264 months), and 66 months after biologics exposure (range 10–105 months) in patients with rheumatoid arthritis [19]. However, the detailed dosage and duration of maintained steroid were not provided. In our study, HBsAg seroreversion was found in 3 cases, where the median time from the first episode of GC pulse therapy to HBsAg seroreversion was 97 months (range 66–168 months). Most of those patients were presented with hepatitis flare and a surge in HBV DNA level. After timely treatment with antiviral therapy, all cases could be safely managed, with no progression to liver failure. It is worth noting that a large portion of patients received maintained steroid treatment in order to maintain the suppression of underlying inflammatory condition after GC pulse therapy. Up to 341 patients (80.4%) had at least a moderate-dose of maintained oral GC (10 mg prednisone equivalent) and amongst them, a high-dose oral GC (20 mg prednisone equivalent) was used in 201 patients (47.4%). All 3 cases experiencing HBsAg seroreversion had been exposed to a high-dose maintained oral GC. It is thus reasonable to highlight the potential risk of maintained steroid in patients receiving GC pulse therapy. Forty-one cases with concurrent use of rituximab were excluded from our analysis. Rituximab is well known for being associated with both HBV reactivation and HBsAg seroreversion in occult HBV infection. Our prior study indicated a 9% HBV reactivation rate in RA patients with occult HBV infection following rituximab treatment [20]. It remains to be clarified whether pulse steroid and rituximab may exhibit additive effects in HBsAg seroreversion. The results of this study indicate that ultra-high dose GC pulse therapy alone does not result in a drastically increased incidence of HBV reactivation in HBsAg-negative, anti-HBc-positive patients. This observation can be partly explained through a previous study which demonstrated that 100 mg of methylprednisolone per day was sufficient to saturate most of the GC receptors in human cells; while GC effects at higher doses were mediated through nongenomic effects [16]. Hence, it is reasonable to postulate that the risk of HBV reactivation in our population depended largely upon concomitant medications, such as a maintained dose of oral steroid rather than short-term GC pulse therapy. The present study is the first to determine the safety of HBsAg-negative, anti-HBc-positive rheumatic patients undergoing ultra-high dose GC pulse therapy. We recommend proper HBV measures and monitoring when other high-risk medications such as rituximab or high dose oral steroid are used concomitantly. However, this study has some limitations. First, this was a medical records review study. Patient characteristics and rheumatic disease entities were not equally distributed between the study groups. In addition, baseline and repeated HBV viral loads were not available for every participant. Therefore, HBV reactivation could have been underestimated in our study. However, a previous study showed that most patients developed HBsAg seroreversion 12–28 weeks after a surge in HBV DNA, and that the kinetics of HBsAg seroreversion were similar to those of alanine aminotransferases after HBV DNA elevation [13]. Therefore, monitoring liver enzymes and the serum marker of HBsAg may be a practical alternative for detecting HBV reactivation in HBsAg-negative, anti-HBc-positive patients. Second, the enrolled patients were taking a variety of concomitant medications over a lengthy enrollment period of 18 years. We also observed 13 patients with systemic sclerosis receiving glucocorticoid pulse therapy for interstitial lung diseases when anti-fibrosis therapy was not available. Physicians’ prescription patterns and awareness of HBV seroreversion may have changed over time. Third, the sample size of HBV seroreversion was limited, since repeated measurement of HBsAg is not universally recommended for rheumatic patients receiving biological therapy in Taiwan [21]. Therefore, the incidence of HBV seroreversion could have been underestimated. Finally, Taiwan is an HBV-endemic area. Whether our results can be extrapolated to countries with a low HBV prevalence remains unclear. A prospective observational study is still required in order to address this issue. In conclusion, amongst HBsAg-negative/anti-HBc-positive rheumatic patients who had received GC pulse therapy, HBV-associated hepatitis was not observed within the first year after GC pulse therapy. A small portion of patients may develop HBsAg seroreversion, but only in those with concomitant high-dose oral steroid use.

## Figures and Tables

**Figure 1 jcm-10-03296-f001:**
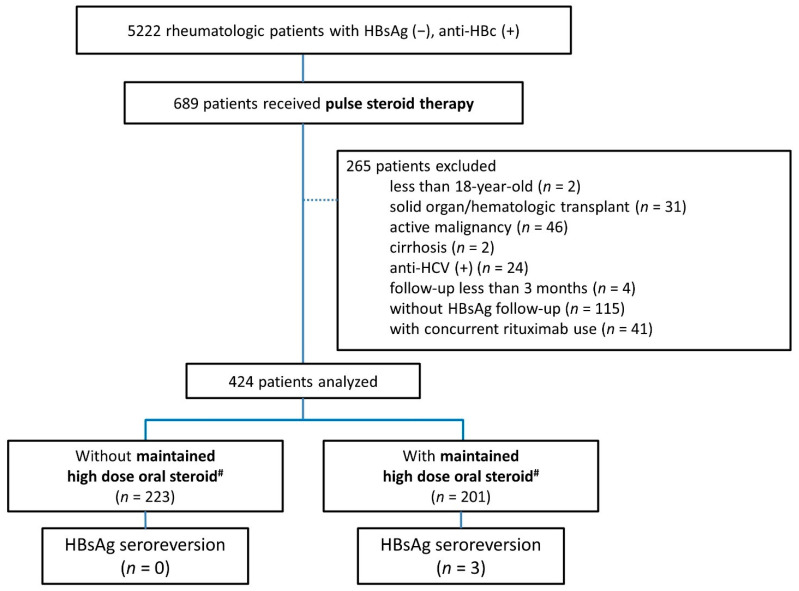
Flowchart of participant recruitments and risk stratification of HBV seroreversion according to concomitant high-risk medications. # prednisolone equivalent dose ≥ 20 mg per day; HBsAg, hepatitis B surface antigen; anti-HBc, antibody to hepatitis B core antigen; HBsAg seroreversion, reappearance of HBsAg from negative to positive; maintained high dose oral steroid, oral prednisolone equivalent over 20 mg/day for over 28 days; anti-HCV, antibody to hepatitis C virus.

**Table 1 jcm-10-03296-t001:** Demographic characteristics of HBsAg-negative and anti-HBc-positive rheumatic patients receiving glucocorticoid pulse therapy.

Characteristics	*N* = 424	
Age (years)	47.0	(37.0–57.1)
Gender (M/F)	126/298	
Follow-up time (months)	131.7	(79.0–170.3)
Platelet (×10^3^/L)	266.5	(207.3–342.3)
Albumin (mg/dL)	3.45	(3–3.9)
Total bilirubin (mg/dL)	0.4	(0.3–0.7)
ALT at baseline, U/L	20	(15–35)
Anti-HBs (+) (*n* = 371)	311	(83.8%)
Number of pulse steroid episodes	2	(1–3)
Underlying rheumatic disease		
Rheumatoid arthritis	140	(33.0%)
Systemic lupus erythematosus	133	(31.4%)
Idiopathic inflammatory myopathy	50	(11.8%)
Ankylosing arthritis	42	(9.9%)
Psoriatic arthritis	18	(4.3%)
Adult onset Still’s disease	15	(3.5%)
Systemic sclerosis	13	(3.1%)
Vasculitis	8	(1.9%)
Sjogren’s syndrome	5	(1.2%)
Concurrent anti-rheumatic treatments		
Anti-TNF-α		
Etanercept	11	(2.6%)
Adalimumab	23	(5.4%)
Golimumab	2	(0.5%)
Abatacept	3	(0.7%)
Maintained oral glucocorticoid ^†^		
10–19 mg/day	140	(33.0%)
≥20 mg/day	201	(47.4%)
DMARDs		
Methotrexate	232	(54.7%)
Sulfasalazine	165	(38.9%)
Azathioprine	152	(35.9%)
Cyclosporine	142	(33.5%)
Leflunomide	49	(11.6%)
Mycophenolate	34	(8.0%)
Cyclophosphamide	98	(23.11%)

^†^ prednisolone equivalent doses for more than 4 weeks. Continuous data were expressed by median (IQR). Categorical data were expressed by number and percentage. HBsAg, hepatitis B surface antigen; anti-HBc, antibody to hepatitis B core antigen; anti-HBs, antibody to hepatitis B surface antigen; ALT, alanine aminotransferase; anti-TNF-α, anti-tumor necrosis factor-alpha; maintained oral prednisolone equivalent, oral prednisolone over 28 days; DMARDs, disease-modifying antirheumatic drugs.

**Table 2 jcm-10-03296-t002:** Demographic characteristics of HBsAg-negative and anti-HBc-positive rheumatic patients receiving glucocorticoid pulse therapy with or without hepatitis within 1st year.

	Hepatitis (+) (*n* = 28)		Hepatitis (−) (*n* = 396)		*p* Value
Age (years)	55.22	(43.59–61.7)	46.41	(36.98–56.55)	0.047 *
Gender (M/F)	12/16		114/282		0.174
Platelet (×103/L)	246	(202–330)	267	(208–353)	0.351
Albumin (mg/dL)	3.3	(2.6–3.7)	3.5	(3.05–3.9)	0.222
Total bilirubin (mg/dL)	0.7	(0.4–1.2)	0.4	(0.3–0.6)	0.006 **
ALT at baseline, U/L	28.5	(16–42.25)	20	(15–33.5)	0.189
Anti-HBs (+) (*n* = 411)	20	(76.92%)	291	(84.35%)	0.404
Anti-HBs titer (*n* = 89)	70.1	(7.3–635.9)	51.9	(13.0–659.3)	0.819
Time to hepatitis (days)	84.8	(17.2–247.3)			
Peak ALT when hepatitis, U/L	157.0	(124.5–206.3)			
Peak ALP when hepatitis, U/L	86	(119–140)			
Seroreversion when hepatitis	0	(0%)			
HBV DNA detected when hepatitis (*n* = 27)	0	(0%)			
Number of pulse steroid episodes	2.5	(1–4)	2	(1–3)	0.490
Underlying rheumatic disease					0.027 *
Rheumatoid arthritis	4	(14.29%)	136	(34.34%)	
Systemic lupus erythematosus	9	(32.14%)	124	(31.31%)	
Idiopathic inflammatory myopathy	8	(28.57%)	42	(10.61%)	
Ankylosing arthritis	1	(3.57%)	41	(10.35%)	
Psoriatic arthritis	3	(10.71%)	15	(3.79%)	
Adult onset Still’s disease	2	(7.14%)	13	(3.28%)	
Systemic sclerosis	0	(0%)	13	(3.28%)	
Vasculitis	1	(3.57%)	7	(1.77%)	
Sjogren’s syndrome	0	(0%)	5	(1.26%)	
Concurrent anti-rheumatic treatments					
Anti-TNF-α					
Adalimumab ^f^	1	(3.57%)	10	(2.53%)	0.533
Etanercept ^f^	1	(3.57%)	22	(5.56%)	1.000
Golimumab ^f^	0	(0%)	2	(0.51%)	1.000
Abatacept ^f^	0	(0%)	3	(0.76%)	1.000
Maintained oral glucocorticoid doses ^†^					1.000
10–19 mg/day	10	(41.67%)	130	(41.01%)	
≥20 mg/day	14	(58.33%)	187	(58.99%)	
DMARDs					
Methotrexate	8	(28.57%)	224	(56.57%)	0.007 **
Azathioprine	16	(57.14%)	136	(34.34%)	0.026 *
Leflunomide ^f^	4	(14.29%)	45	(11.36%)	0.550
Cyclosporine	12	(42.86%)	130	(32.83%)	0.379
Sulfasalazine	4	(14.29%)	161	(40.66%)	0.010 *
Mycophenolate ^f^	6	(21.43%)	28	(7.07%)	0.018 *
Cyclophasphamide	11	(39.29%)	87	(21.97%)	0.062
Cyclophosphamide	11	(34.38%)	96	(22.17%)	0.172

Chi-Square test. ^f^ Fisher’s Exact test. * *p* < 0.05, ** *p* < 0.01. ^†^ prednisolone equivalent doses. Continuous data were expressed by median (IQR). Categorical data were expressed by number and percentage. HBsAg, hepatitis B surface antigen; anti-HBc, antibody to hepatitis B core antigen; anti-HBs, antibody to hepatitis B surface antigen; ALT, alanine aminotransferase; ALP, Alkaline phosphatase; anti-TNF-α, anti-tumor necrosis factor-alpha; maintained oral prednisolone equivalent, oral prednisolone over 28 days; DMARDs, disease-modifying antirheumatic drugs.

**Table 3 jcm-10-03296-t003:** Risk factors associated with hepatitis after receiving glucocorticoid pulse therapy in HBsAg-negative, anti-HBc-positive rheumatic patients.

	Univariate Model			Multivariate Model		
	OR	(95% CI)	*p* Value	OR	(95% CI)	*p* Value
Age	1.03	(1.00–1.05)	0.069	1.04	(1.01–1.07)	0.021 *
Concurrent anti-rheumatic treatments						
Maintained oral glucocorticoid ^†^						
10–19 mg/day	Reference					
≥20 mg/day	0.97	(0.42–2.26)	0.950			
DMARDs						
Methotrexate	0.31	(0.13–0.71)	0.006 **	0.54	(0.21–1.42)	0.211
Azathioprine	2.55	(1.17–5.54)	0.018 *	2.50	(1.03–6.06)	0.043 *
Leflunomide	1.30	(0.43–3.92)	0.641			
Cyclosporine	1.53	(0.71–3.34)	0.280			
Sulfasalazine	0.24	(0.08–0.71)	0.010 *	0.42	(0.12–1.43)	0.165
Mycophenolate	3.58	(1.34–9.56)	0.011 *	2.83	(0.94–8.52)	0.064
Cyclophosphamide	2.30	(1.04–5.09)	0.040 *	1.02	(0.41–2.54)	0.963

* *p* < 0.05, ** *p* < 0.01. ^†^ oral prednisolone equivalent doses. OR, odds ratio; 95% CI, 95% confidence interval; HBsAg, hepatitis B surface antigen; anti-HBc, antibody to hepatitis B core antigen; maintained oral prednisolone equivalent, oral prednisolone over 28 days; DMARDs, disease-modifying anti-rheumatic drugs.

**Table 4 jcm-10-03296-t004:** Clinical, serological and virological characteristics of rheumatic patients with HBsAg seroreversion after receiving glucocorticoid pulse therapy.

Characteristics	Patient 1	Patient 2	Patient 3
Age/Gender	66/F	25/F	59/F
Diagnosis	Idiopathic Inflammatory myopathy	SLE	RA
Baseline anti-HBs	Positive	Negative	Positive
anti-HBs loss at seroreversion	yes	N/A	yes
Baseline ALT (U/L)	17	36	30
Peak ALT at seroreversion (U/L)	26	694	71
Baseline total Bilirubin (mg/dL)	0.3	0.3	0.4
Peak Bilirubin at seroreversion (mg/dL)	0.4	0.1	0.4
Viral loads at baseline (IU/mL)	N/A	3.48 × 10^2^	Undetectable
Viral loads at reactivation (IU/mL)	1.03 × 10^8^	>1.7 × 10^8^	8.79 × 10^7^
Concomitant drugs			
Anti-TNF-α biologics			
Maintained oral glucocorticoid ≥20 mg/day ^†^	+	+	+
Time to seroreversion (months)	97	66	168
NUCs usage at seroreversion	ETV	ETV	ETV
Outcomes	Stable	Stable	Stable

^†^ oral prednisolone equivalent doses. HBsAg, hepatitis B surface antigen; anti-HBc, antibody to hepatitis B core antigen; anti-HBs, antibody to hepatitis B surface antigen; HBsAg seroreversion, reappearance of HBsAg in anti-HBc-positive patients; ALT, alanine aminotransferase; anti-TNF-α, anti-tumor necrosis factor-alpha; NUC, nucleoside analogue; ETA, Etanercept; ETV, Entecavir; M, male; F, female; RA, rheumatoid arthritis; SLE, systemic lupus erythematosus; N/A, Not applicable.

## Data Availability

Data available on request from the authors.

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
