# Peer review of "Long-Term Safety in HBsAg-Negative, HBcAb-Positive Patients with Rheumatic Diseases Receiving Maintained Steroid Therapy after Pulse Therapy"

_jcm, 2021, doi:10.3390/jcm10153296_

Round 1

Reviewer 1 Report

This is a well performed study and a well written masnuscript describing al low percentage of patients with variuos rheumatic conditions developping hepatopathy after GC pulse therapy, some of which receiving higher oral GC doses for several weeks. The study shows, that this treatment is safe and does not bear a relevant risk in terms of reactivation of HBV-infection

From a rheumatological point of view, it would be interesting to know why the patients were selected for GC pulse therapy , most likely because of clinical and/or humoral activity/relapse of the underlying disease; however, in patients with systemic sclerosis GC pulse therapy is not a well established therapeutic options; maybe this can briefly be clarified (a short explanation would be good enough)

Author Response

Thanks for your excellent review and useful comments

This is a well performed study and a well written masnuscript describing al low percentage of patients with variuos rheumatic conditions developping hepatopathy after GC pulse therapy, some of which receiving higher oral GC doses for several weeks. The study shows, that this treatment is safe and does not bear a relevant risk in terms of reactivation of HBV-infection

From a rheumatological point of view, it would be interesting to know why the patients were selected for GC pulse therapy , most likely because of clinical and/or humoral activity/relapse of the underlying disease; however, in patients with systemic sclerosis GC pulse therapy is not a well established therapeutic options; maybe this can briefly be clarified (a short explanation would be good enough)

We agree with the reviewer’s comment that glucocorticoid pulse therapy is not currently the treatment of choice. Since the study enrolled participants during the period of 2006 to 2018, when anti-fibrosis therapy was not available, the majority of patients with systemic sclerosis received glucocorticoid pulse therapy due to interstitial lung disease. We added this in the discussion section (page 11, lines 292-293). Thank you for your valuable input.

Reviewer 2 Report

1) General comments

Dr. Lin, Dr. Yang and Dr. Chen, et al. reported "Long-term Safety in HBsAg-negative, HBcAb-positive Patients with Rheumatic Diseases After Receiving Glucocorticoid Pulse Therapy". The article is informative and well presented. The reviewer has some comments.

  1. The reviewer would like to confirm in page 5, line 151, ‘the cause of hepatitis was mostly drug-induced’. Please explain how the authors diagnosed drug-induced hepatitis after GC pulse therapy in Discussions.

  1. The reviewer thinks the authors investigated the ratio of HBsAg seroreversion receiving maintained steroid therapy after pulse therapy. And the authors described ‘The median time 186 from the first episode of GC pulse therapy to HBsAg seroreversion was 97 months (range 187 66-168 months)’. The reviewer thinks this ratio of HBsAg seroreversion in this study was not shown directly effects of pulse therapy. Please reconfirm the definitions of pulse therapy in Materiala and Methods.

  1. The reviewer recommends the consideration of changing the title ‘Long-term Safety in HBsAg-negative, HBcAb-positive Patients with Rheumatic Diseases After Receiving Glucocorticoid Pulse Therapy’ to ‘Long-term Safety in HBsAg-negative, HBcAb-positive Patients with Rheumatic Diseases Receiving Maintained Steroid Therapy After Pulse Therapy’.

Author Response

Reply to the comments and suggestions from Reviewer 2

Thanks for your excellent review and useful comments

Dr. Lin, Dr. Yang and Dr. Chen, et al. reported "Long-term Safety in HBsAg-negative, HBcAb-positive Patients with Rheumatic Diseases After Receiving Glucocorticoid Pulse Therapy". The article is informative and well presented. The reviewer has some comments.

  1. The reviewer would like to confirm in page 5, line 151, ‘the cause of hepatitis was mostly drug-induced’. Please explain how the authors diagnosed drug-induced hepatitis after GC pulse therapy in Discussions.

Thanks for the reviewer’s comments. We classified patients with drug-induced hepatitis on the basis of temporal relationship and the exclusion of alternative explanations, ie. acute viral hepatitis other than HBV, alcoholic hepatitis or infectious liver diseases [11]. We have revsied both the method (page 4, lines 131-133) and discussion sections (page 9, lines 226-228).

  1. The reviewer thinks the authors investigated the ratio of HBsAg seroreversion receiving maintained steroid therapy after pulse therapy. And the authors described ‘The median time 186 from the first episode of GC pulse therapy to HBsAg seroreversion was 97 months (range 187 66-168 months)’. The reviewer thinks this ratio of HBsAg seroreversion in this study was not shown directly effects of pulse therapy. Please reconfirm the definitions of pulse therapy in Material and Methods.

  Thanks for the reviewer’s comments. We agree with reviewer’s comment that HBsAg seroreversion may not be directly attributed to pulse therapy. A recent study had also shown that HBsAg seroreversion can occur 120 months after steroid exposure (range 20-264 months), and 66 months after biologics exposure (range 10-105 months) in patients with rheumatoid arthritis [19] (Clin Gastroenterol Hepatol. 2020, 18, 2573-2581). Although the definite cause of HBsAg seroreversion has not been clearly established in the available literature or in our study, it may be associated with the cumulative effects of steroid or other immunosuppressants. In our study, we believe that maintained steroid above 20mg for at least 4 weeks after pulse therapy may be associated with HBsAg seroreversion. We revised the definitions of glucocorticoid therapy in Material and Methods (page 4, lines 112-119) and discussed this point in the discussion section (page 10, lines 249-251).

  1. The reviewer recommends the consideration of changing the title ‘Long-term Safety in HBsAg-negative, HBcAb-positive Patients with Rheumatic Diseases After Receiving Glucocorticoid Pulse Therapy’ to ‘Long-term Safety in HBsAg-negative, HBcAb-positive Patients with Rheumatic Diseases Receiving Maintained Steroid Therapy After Pulse Therapy’.

Thanks for the reviewer’s comments. We revsied the title to ‘Long-term Safety in HBsAg-negative, HBcAb-positive Patients with Rheumatic Diseases Receiving Maintained Steroid Therapy After Pulse Therapy’.

Round 2

Reviewer 2 Report

1) General comments

Dr. Lin, Dr. Yang and Dr. Chen, et al. revised "Long-term Safety in HBsAg-negative, HBcAb-positive Patients with Rheumatic Diseases Receiving Maintained Steroid Therapy After Pulse Therapy". The article is informative and well presented. The reviewer has no comments.